# Toward Robust Spiking Neural Network Against Adversarial Perturbation

**Ling Liang**
UC Santa Barbara
lingliang@ucsb.edu

**Kaidi Xu**
Drexel University
kx46@drexel.edu

**Xing Hu**
SKL of Processors
Institute of Computing Technology, CAS
huxing@ict.ac.cn

**Lei Deng**
Tsinghua University
leideng@mail.tsinghua.edu.cn

**Yuan Xie**
Alibaba Group
yuanxie@gmail.edu

## Abstract

As spiking neural networks (SNNs) are deployed increasingly in real-world efficiency critical applications, the security concerns in SNNs attract more attention. Currently, researchers have already demonstrated an SNN can be attacked with adversarial examples. How to build a robust SNN becomes an urgent issue. Recently, many studies apply certified training in artificial neural networks (ANNs), which can improve the robustness of an NN model promisely. However, existing certifications cannot transfer to SNNs directly because of the distinct neuron behavior and input formats for SNNs. In this work, we first design S-IBP and S-CROWN that tackle the non-linear functions in SNNs' neuron modeling. Then, we formalize the boundaries for both digital and spike inputs. Finally, we demonstrate the efficiency of our proposed robust training method in different datasets and model architectures. Based on our experiment, we can achieve a maximum 37.7% attack error reduction with 3.7% original accuracy loss. To the best of our knowledge, this is the first analysis on robust training of SNNs.

## 1 Introduction

Spiking neural networks (SNNs) are a class of models that have the potential ability to simulate the behavior of neuron circuits in the brain. Among the rich biological characters of SNNs, the event-driven information propagation mechanism makes SNNs easy to deploy with limited computation resources. Many neuromorphic chips are designed to achieve low-power SNN inference, such as TrueNorth [1], Loihi[8], SpinalFlow [23] and H2Learn [20].

With more attention to the study of SNNs, the security issues also raise concerns in the community. Adversarial attack [27, 7, 2, 34] is one of the most intuitive ways to evaluate the robustness of a model. In adversarial attacks, the attacker generates adversarial examples to fool a model that has wrong prediction. Currently, SNNs are demonstrated can be attacked through adversarial examples [26, 19, 6, 21]. It is urgent to explore an efficient way to improve the robustness of SNN models.

Previously, researchers have investigated the impact of hyperparameter selection [10] and input filtering [21] on the adversarial attack in SNNs. However, these methods do not directly promote the classification behavior of a given SNN model. [5] proposes a robust training method that relies on the adversarial attack in parameter space. Unlike SNNs, how to improve the robustness of an artificial neural network (ANN) has been well studied. Recently, trianing a neural network model with certified defense methods [30, 22, 38, 33] show remarkable guarantee to improve the model's robustness. CROWN-IBP [38] is one of the most promising certified training methods with

polynomial computational cost compared to natural training. The CROWN-IBP method computes the output boundary for a given bounded input. However, the current CROWN-IBP method cannot be applied directly to SNNs. Firstly, the neuron dynamic in SNNs is more complicated. Hence, new boundary functions should be defined. Secondly, SNNs accept both spike and digital inputs, which requires additional boundary generalization for different input types.

Enlightened by certification training in ANNs, we designed an end-to-end robust training method to improve the robustness of an SNN model against adversarial attacks. Specifically, our contributions can be summarized as follows:

- We design S-IBP and S-CROWN to tackle the non-linear fire function and temporal update in SNNs which firstly introduce SNNs to the linear relaxation based verification family.
- We formalized $\ell_0$-norm and $\ell_\infty$-norm boundaries for spike and digital inputs, respectively.
- Our proposed methods are evaluated on MNIST [18], FMNIST [32] and NMNIST [24] datasets[1]. The experimental results show that we can achieve a maximum $37.7\%$ attack error reduction with $3.7\%$ original accuracy loss.

## 2 Preliminary and Related Work

### 2.1 Spiking Neural Netowks (SNNs)

**Neuron Modeling**: Usually, SNNs are designed to simulate the neuron behavior in the brain. In this work, we adopt the well-studied leaky integrated-and-fire (LIF) [12] for the neuron modeling as Figure 1(a). The state of a neuron at each time step is determined by its membrane potential $m$ and its spike status $s$. Specifically, a neuron will $fire$ a spike to the post-synaptic neurons once its membrane potential is greater than a threshold $th$ and its membrane potential will be reset to 0 at the same time. An example of a neuron dynamic is shown in Figure 1(b) which simulates a neuron's behavior in 5 time steps. The spike status of a neuron can be formulated as

$$s_t[k] = fire(m_t[k] - th). \tag{1}$$

$$fire(x) = \begin{cases} 1, x \geq 0 \\ 0, \text{ otherwise.} \end{cases} \tag{2}$$

We use $m_t[k]$ and $s_t[k]$ to represent neurons' membrane potentials and spike status at $t$-th time step and $k$-th layer. $fire(\cdot)$ is the Heaviside step function. The membrane potential of a neuron is composed of spatial and temporal update that follows

$$m_t[k] = \underbrace{\sum_t w[k-1] \otimes s_t[k-1]}_{spatial} + \underbrace{\alpha m_{t-1}[k] \cdot (1 - s_{t-1}[k])}_{temporal}. \tag{3}$$

The spatial update is a Convolution or Fully Connected or Pooling operation between the weight and the spike events in previous layer. The temporal update is determined by the neuron's membrane potential and spike status in the previous time step. Here, $\alpha$ is a scaling factor to control the decay.

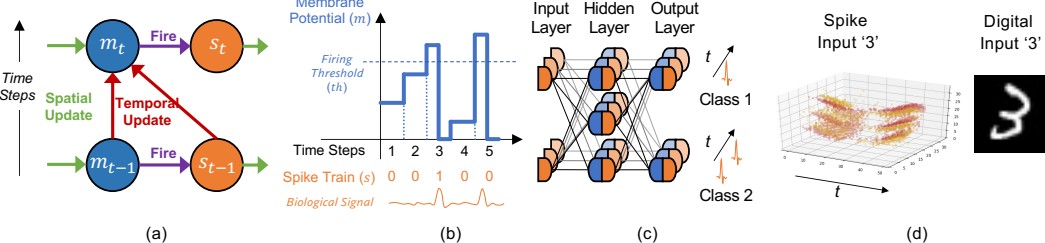

Figure 1: Introduction of SNNs: (a) neuron modeling; (b) computing model of a neuron; (c) spatiotemporal flow of an SNN model; (d) input formats of an SNN.

**Network Structure**: SNNs are built with an input layer, hidden layers, and an output layer as Figure 1(c). In SNNs, the input layer receives spike inputs. The hidden layers are CONV/FC/POOL that correspond to the spatial update in the neuron modeling. Compared to ANNs, SNNs involve an

---
[1]https://github.com/liangling76/certify_snn

additional temporal axis that passes the neurons' status along with time steps. The final recognition of an SNN is determined by the spike events of the output layer. In this work, we adopt the rate coding, i.e., the neuron that fires the most spikes indicates the classification result.

**Input Format**: In this work, we focus on image recognition tasks. The input of an SNN can be spike events captured by DVS devices [25]. Also, SNNs can take a digital image as input with an additional sampling. In this work, we adopt Bernoulli sampling [9] for digital inputs that follows

$$P(\dot{x}_t[i] = 1) = \hat{x}[i]. \tag{4}$$

We use $\dot{x}_t$ to represent the spike input at time step $t$. $\hat{x}$ is the digital input after normalization to $[0, 1]$. The example of spike and digital inputs are shown in Figure 1 (d).

**Training Method**: Recently, supervised learning via backpropagation through time (BPTT) has been adopted in SNN training [31]. The BPTT-based training method can achieve a higher accuracy and scalability compared to unsupervised learning algorithms. Also, the SNNs trained with BPTT require fewer time steps, which is more hardware friendly in the real deployment. Thus, in this work, we focus on the SNNs trained with the BPTT-based learning method.

## 2.2 Adversarial Attack in SNN

The generation of adversarial examples is one of the most powerful ways of affecting a model's robustness [27, 7, 39, 36]. In this work, we use adversarial attacks as an assessment tool to evaluate the robustness of an SNN model. Specifically, in this work we adopt a gradient-based attack method [19] for SNNs.

**Untargeted White-box Gradient-based Attack**: In adversarial attacks, the attacker adds imperceptible noise on inputs to fool the model to predict wrong results. In untargeted attack, the model predicts the adversarial example as any other classes except the ground-truth, which can be formulated as

$$\underset{\delta}{\arg\min} \|\delta\|_p, \ s.t. \ f(x + \delta) \neq f(x). \tag{5}$$

Here, $f(\cdot)$ is the prediction result, $x$ is the original input, and $\delta$ is the noise added on the input. In this work, we adopt white-box attack, which means the attacker knows all of the model information.

The gradient-based attacks are the most efficient ways to generate adversarial examples. In SNNs, the gradient-based attack for the spike and digital inputs can be formulated as

$$\begin{cases} \dot{x}'_{n+1} = \dot{x}'_n + F[\nabla_{\dot{x}'_n} L(\theta, \dot{x}'_n, y_{org})], & \text{spike input,} \\ \\ \hat{x}'_{n+1} = \hat{x}'_n + \dfrac{\sum_t F[\nabla_{\dot{x}'_n} L(\theta, \dot{x}'_n, y_{org})]}{T}, & \text{digital input.} \end{cases} \tag{6}$$

We use $\dot{x}'$ and $\hat{x}'$ to represent the spike and digital adversarial examples, i.e., $x' = x + \delta$. $L$ and $\theta$ are the loss function and parameters of the model, respectively. The adversarial example is constructed by adding the gradient of inputs with the original label ($y_{org}$) in loss function. $F$ is a filter function that samples, clips, and generates spike compatible noise. During the attack, attackers can compute the adversarial examples iteratively. We use $n$ to denote the attack iteration.

## 2.3 Certified Training

Recently, the certified training [13, 38, 33] has been demonstrated to improve the guaranteed robustness of a neural network. In this work, we leverage the CROWN-IBP method [38], one of the state-of-the-art certified training that can achieve tighter bounds in accepctable training cost. Considering a digital input data bounded with $\ell_\infty$-norm, the goal of CROWN-IBP is to identify whether arbitrary input data within the boundary can fool the model. The CROWN-IBP contains two parts of bounding methods: IBP [13] and CROWN [37].

**IBP**: In IBP processing, the lower and upper bounds of each layer's feature map are computed along the forward propagation. During the bound computation, the linear operation can be easily bounded once we know the maximum and minimum values of input. However, the upper and lower bounds after the non-linear operations need to be dedicated to analysis. Once we acquire the bounds of output we can evaluate the robustness of the model for the given input purbation set.

**CROWN**: Unlike IBP, CROWN bounds the model in a backward propagation manner recursively. The goal of CROWN is to formulate the output bounds as a linear equation of input. In order to achieve this goal, every operation in an NN model should be bounded by two linear equations.

Although CROWN can achieve very tight bounds, the computational cost is remarkably higher than IBP. So CROWN-IBP, by combining the fast IBP bounds in a forward bounding pass and CROWN in a backward bounding pass can efficiently and consistently outperforms IBP baselines on training verifiably robust neural networks.

**Integrate with Other Approaches**: One of the promising characteristics of certified training is that it can be integrated with other algorithms. For example, the combination of certified training and adversarial training can further improve the robustness of a model [3, 11]. For the operations which can improve the performance of a model and only involve linear operations, the certified training algorithm can be directly applied without causing additional relaxation loss [28, 29]. Currently, the adversarial training [17] and batch normalization [14, 15, 4] have demonstrated their effectiveness in improving the robustness of an SNN model, which have the potential to be integrated into our proposed design in the future.

## 3 Methodology

### 3.1 S-IBP & S-CROWN

As described in Section 2.3, the core mission of certification training is to find the bounds of IBP and CROWN for each function. The information propagation in SNNs includes $fire$, temporal update and spatial update as shown in Figure 1(a). In this subsection, we detail the upper bound and lower bound of S-IBP and S-CROWN for each function. During the bound formulation of a function, we assume the S-IBP bounds for function inputs have been acquired.

**Fire**: The $fire$ function describes the relationship between the membrane potential $m_t$ and spike $s_t$ of a neuron as Equation 1 and 2, i.e. once the neuron's membrane potential is greater than a threshold $th$, the neuron will fire a spike. Assume that we have already acquired the S-IBP upper bound $m_t^u$ and lower bound $m_t^l$ for membrane potential, the S-IBP bounds for spike can be calculated with

$$s_t^u = fire(m_t^u - th), \ \ s_t^l = fire(m_t^l - th). \tag{7}$$

During S-CROWN, we need to find two linear equations to bound the $fire$ function. When the S-IBP upper bound of membrane potential $m_t^u$ is smaller than the threshold $th$, we can conclude that the neuron must not fire. Also, when the S-IBP lower bound $m_t^l$ is greater than the threshold, we can make sure the neuron must fire. Thus, we mainly need to consider the unstable case, i.e., $m_t^l < th \leq m_t^u$. In order to achieve the lowest bound relaxation, we design two boundary systems as Figure 2(a) and (b). We use the red line to represent the $fire$ function. The blue and yellow lines represent the boundary functions. When the S-IBP lower bound of membrane potential $m_t^l$ is much smaller than $th$, we set the S-CROWN lower bound to $s_t = 0$ and the S-CROWN upper bound is a line that crosses $(m_t^l, 0)$ and $(th, 1)$. In contrast, when $m_t^u$ is much larger than $th$, we set the S-CROWN upper bound to $s_t = 1$ and the S-CROWN lower bound is a line that passes $(th, 0)$ and $(m_t^u, 1)$. The bound can be potentially optimized by solving the decent gradient [35]. Overall, the S-CROWN boundary for the $fire$ function under different cases can be summarized as

$$\begin{cases} 0 \leq s_t \leq 0, & m_t^u < th, \\ 1 \leq s_t \leq 1, & m_t^l \geq th, \\ 0 \leq s_t \leq \frac{m_t - m_t^l}{th - m_t^l}, & 0 \leq m_t^u - th < th - m_t^l \\ \frac{m_t - th}{m_t^u - th} \leq s_t \leq 1, & m_t^u - th \geq th - m_t^l > 0 \end{cases} \tag{8}$$

**Temporal Update**: According to Equation 3, the temporal update of membrane potential can be formalized as $\alpha m_t(1 - s_t)$. Since $\alpha$ is a constant factor, in this subsection we do not include it in the boundary analysis. Assume $z_t = m_t(1 - s_t)$, and the S-IBP upper and lower bounds for membrane potential at time step $t$ has been acquired. The relation between $m_t$ and $z_t$ can be described through Figure 2(c). Based on the LIF model, the membrane potential will be reset to 0 once its value acrosses

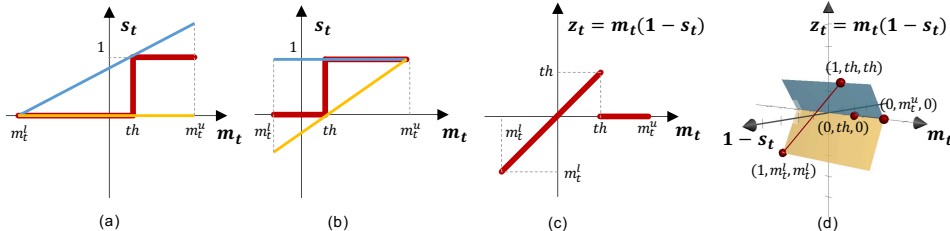

(a)           (b)           (c)           (d)

Figure 2: S-CROWN upper and lower bounds for the $fire$ function when (a) $m_t^l$ is much smaller than $th$ and (b) $m_t^u$ is much larger than $th$. (c) temporal part of memory potential update. (d) S-CROWN upper and lower bounds for the temporal part of memory potential update (unstable case).

the pre-defined threshold $th$. Thus, the S-IBP bounds for $z_t$ can be computed with

$$
\begin{cases}
z_t^u = m_t^u, \ z_t^l = m_t^l, & m_t^u < th, \\
z_t^u = z_t^l = 0, & m_t^l \geq th, \\
z_t^u = th, \ z_t^l = min(0, m_t^l), & m_t^l < th \leq m_t^u.
\end{cases}
\tag{9}
$$

When $m_t^u < th$ or $m_t^l \geq th$, there is no boundary relaxation in S-IBP. We only need to consider the boundary when $m_t^l < th \leq m_t^u$ (unstable case), where $z_t \in [min(0, m_t^l), th)$ for unstable case.

During S-CROWN, $z_t$ can be also bounded without relaxation when $m_t^u < th$ and $m_t^l \geq th$. The S-CROWN upper bound (blue plane) and lower bound (yellow plane) for $z_t$ when $m_t^l < th \leq m_t^u$ are shown in Figure 2(d). Here, we use an additional $(1 - s_t)$ axis to help us build the boundary. Note that the membrane potential is related to the spike status. Once $s_t = 1 \rightarrow (1 - s_t) = 0$, $m_t$ must be greater than $th$ and $z_t = 0$. When $s_t = 0 \rightarrow (1 - s_t) = 1$, $m_t$ must be smaller than $th$ and $z_t = m_t$. Thus, we need to find two planes to bound these two functions (red lines in Figure 2(d)). In summary, the S-CROWN boundary for the temporal update can be formulated as

$$
\begin{cases}
m_t \leq z_t \leq m_t, & m_t^u < th, \\
0 \ \leq z_t \leq 0, & m_t^l \geq th, \\
(1 - s_t)m_t^l < z_t < (1 - s_t)th, & m_t^l < th \leq m_t^u.
\end{cases}
\tag{10}
$$

**Spatial Update**

The spatial update in SNNs is shown in Equation 3. Similar to ANNs, the spatial update in SNNs is composed of CONV/FC/POOL (in this work we focus on the CONV and FC). In SNNs CONV/FC takes spike events and weight as input. Since the spike events are in binary format, the S-IBP of spatial update can be implemented with

$$
\begin{cases}
center = w[k] \otimes s_t^l[k] + b[k], \\
sp_t^u[k + 1] = center + w[k]^+ \otimes (s_t^l[k] = 0 \cap s_t^u[k] = 1), \\
sp_t^l[k + 1] \ = center + w[k]^- \otimes (s_t^l[k] = 0 \cap s_t^u[k] = 1).
\end{cases}
\tag{11}
$$

We use $sp_t$ to represent the result of the spatial update. $\otimes$ denotes CONV/FC operation. In Equation 11, $s_t^l[k] = 1$ represents those pre-synaptic neurons in layer $k$ who are must fire. The stable fired pre-synaptic neurons contribute the same for both S-IBP upper and lower bound of $sp_t$. The unstabled pre-synaptic neurons can be represented with $(s_t^l[k] = 0 \cap s_t^u[k] = 1)$, whose upper and lower bound for spike status are 1 and 0. These unstable pre-synaptic neurons will perform CONV/FC with the positive and negative weights to affect the S-IBP upper and lower bound of $sp_t$.

Since the spatial update is a linear operation, the S-CRWON for spatial update does not have relaxation. We do not need to design boundary functions to bound the spatial update in SNNs.

## 3.2 Input Boundary Formalization

In SNNs, the input layer of an SNN model only accepts binary spike. The special data format for the input layer leads to different boundary formalizations for spike and digital images.

**Spike Input**: An example of spike input is shown in Figure 1(d). All elements in a spike data are in binary format. For each element $\dot{x}_t[i]$ in a spike input, the boundary of that element can be either

stable cases: $\dot{x}_t^u[i] = \dot{x}_t^l[i] = 0$; $\dot{x}_t^u[i] = \dot{x}_t^l[i] = 1$ or unstable case: $\dot{x}_t^u[i] = 1, \dot{x}_t^l[i] = 0$. For a spike input with uncertainty noise, we can only control how many data points in the spike input are unstable. Thus, the boundary for a spike input can be formulated with $\ell_0$-norm. Specifically, we can pick $size(\dot{x}) \times \epsilon$ elements from a spike input and set them as unstable points. Also, the $\ell_0$-norm boundary can be interpreted as the probability of an unstable element, which can be formulated as

$$P(\dot{x}_t^u[i] = 1, \dot{x}_t^l[i] = 0) = \epsilon \iff |\dot{x}' - \dot{x}|_0 \leq size(\dot{x}) \times \epsilon \tag{12}$$

Here, $\dot{x}'$ is an arbitrary adversarial example that has at most $size(\dot{x}) \times \epsilon$ data points different from $\dot{x}$.

Note that we can only certify the robustness of a spike input after we have picked the unstable data points. Under our robustness formulation, we cannot guarantee the robustness of a spike input under a given $\ell_0$-norm. The reason is that the search space for $\ell_0$-norm cannot be bounded.

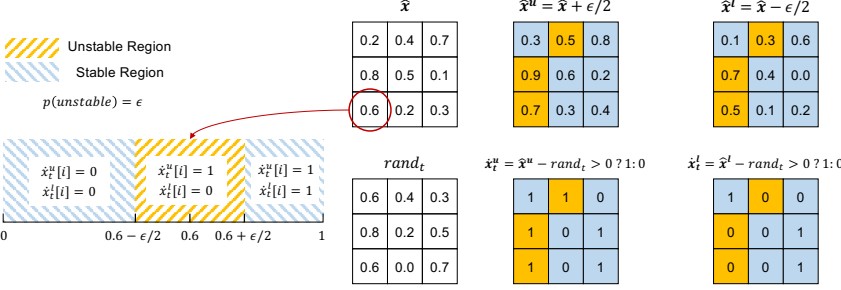

Figure 3: Relation between $\ell_\infty$-norm digital input doundary and $\ell_0$-norm spike input boundary. In this example, $\epsilon = 0.2$.

**Digital Input**:

Digital inputs need an additional sampling before feeding the data to the input layer as Equation 4. Note that we have defined the $\ell_0$-norm boundary for a spike input. We need to further explore how to define a boundary for a digital input when its sampled data is bounded with $\ell_0$-norm. Based on our analysis, we find that the corresponding digital input boundary can be formalized with $\ell_\infty$-norm as Figure 3. We use $rand_t$ to represent a rand mask to achieve Bernoulli sampling for each time step:

$$\dot{x}_t[i] = \begin{cases} 1, & \hat{x}[i] > rand_t[i], \\ 0, & \text{otherwise.} \end{cases} \tag{13}$$

After we bound the digital input with $\ell_\infty$-norm, the upper bound of digital image becomes $\hat{x}^u = \hat{x} + \epsilon/2$ and the lower bound is $\hat{x}^l = \hat{x} - \epsilon/2$. For all elements in $\hat{x}$, the difference between the upper bound and the lower bound is $\epsilon$, i.e., $\hat{x}^u[i] - \hat{x}^l[i] = \epsilon$. For each time step, $\hat{x}^u$ and $\hat{x}^l$ use the identical random map $rand_t$ to sample the corresponding spike inputs $\dot{x}_t^u$ and $\dot{x}_t^l$. Based on the sampling processing, the probability of an element in input layer is unstable can be formulated as

$$P(\dot{x}_t^u[i] = 1, \dot{x}_t^l[i] = 0) = \hat{x}^u[i] - \hat{x}^l[i] = \epsilon. \tag{14}$$

From the static perspective, the $\ell_\infty$-norm boundary for digital input is equivalent to the $\ell_0$-norm boundary for spike input.

For digital inputs, although we apply $\ell_\infty$-norm on input boundary, we cannot guarantee the robustness of the input under such boundary. Since for an digital input, the possible sampled results are equivalent to the entire space (each element in the spike input can be 1 or 0), which cannot be bounded.

For both spike and digital inputs, we need to carefully select $\epsilon$ for robust training. An extremely small $\epsilon$ will cause the upper and lower bound identical which does not consistent with linear relaxation. A large $\epsilon$ may cause the following robust training to diverge. In this work, $\epsilon$ is selected based on the noise magnitude in adversarial attacks and some empirical experiments.

### 3.3 Robust Training Algorithm

In this subsection, we present the end-to-end robust training for an SNN. For the overall flow, our method will first run S-IBP to get the boundary of intermediate data. Then we apply the S-CROWN which needs the S-IBP bound. Finally, we apply the certification-based robust training.

---

**Algorithm 1** S-IBP

---

**Input:**

spike input $\dot{x}$ or digital input $\hat{x}$; input boundary $\epsilon$; robust training time steps $T'$;

**Func:**

**for** $t = 1$ **to** $T'$ **do**

    // input boundary formalization

    **if** $\hat{x}$ **then**

        generate random map $rand_t$;

        $\dot{x}_t^u = (\hat{x} + \epsilon/2) - rand_t > 0 ? 1 : 0;$

        $\dot{x}_t^l = (\hat{x} - \epsilon/2) - rand_t > 0 ? 1 : 0;$

    **else**

        Randomly pick $size(\dot{x}_t) \times \epsilon$ elements from $\dot{x}_t$ and label the picked elements with $pick_t$;

        $\dot{x}_t^u = \dot{x}_t^l = \dot{x}_t; \ \ \dot{x}_t^u[pick_t] = 1; \ \ \dot{x}_t^l[pick_t] = 0;$

    **end if**

    $s_t^u[0] = \dot{x}_t^u; \ \ s_t^l[0] = \dot{x}_t^l;$

    inital $m_t^u[k] = m_t^l[k] = 0$ for all layers.

    //S-IBP

    **for** $k = 1$ **to** $K$ **do**

        //spatial update (Eq. (11))

        $m_t^u[k] + = sp_t^u[k]; \ \ m_t^l[k] + = sp_t^l[k]$

        //temporal update (Eq. (9))

        **if** $t < T'$ **then**

            $m_{t+1}^u[k] = \alpha * z_t^u; \ \ m_{t+1}^l[k] = \alpha * z_t^l;$

        **end if**

        //fire (Eq. (7))

        $s_t^u[k] = fire(m_t^u[k] - th); \ \ s_t^l[k] = fire(m_t^l[k] - th);$

    **end for**

**end for**

**Return** upper and lower bound of intermediate data $\dot{x}_t^u, \dot{x}_t^l, m_t^u, m_t^l, s_t^u, s_t^l$

---

**Flexible Time Steps**: We note that for each SNN layer, the parameters are shared among different time steps. Thus, we can set arbitrary time steps $T'$ for the robust training.

**S-IBP**: During the robust training, the inputs first pass the S-IBP as Algorithm 1. At the beginning, the input is bounded according to the input type. Then, the intermediate data are bounded along the forward direction. Finally, the upper and lower bounds of all intermediate data are stored which will be used during the S-CROWN phase.

**S-CROWN**: Note that the goal of S-CROWN is to formulate the output bounds of a model as linear equations of input's upper and lower bounds. The boundary of S-CROWN is computed from the backward direction, and the details are shown in Algorithm 2. Finally, we can follow the robust training method in [38] and use the lower bound of S-CROWN to train the SNN model.

---

**Algorithm 2** S-CROWN

---

**Input:**

$\dot{x}_t^u, \dot{x}_t^l, m_t^u, m_t^l, s_t^u, s_t^l$ from S-IBP;   robust training time steps $T'$;

**Func:**

//S-CROWN

build identity $I$ matrix;   $As_t[K] = I/T';$

**for** $k = K$ **to** $1$ **do**

    initial $Am_t[k] = 0$ for all time steps

    **for** $t = T'$ **to** $1$ **do**

        //fire (Eq. (8))

        build $m_t[k] * d1^l + b1^l \le s_t[k] \le m_t[k] * d1^u + b1^u;$

        $Am_t[k] + = As_t[k]^- * d1^u + As_t[k]^+ * d1^l;$

        $bias \ \ \ + = As_t[k]^- * b1^u + As_t[k]^+ * b1^l;$

        //temporal update (Eq. (10))

        **if** $t > 1$ **then**

            build $m_{t-1}[k] * d2^l + (1 - s_{t-1}[k]) * d3^l \le z_{t-1}[k];$

            build $m_{t-1}[k] * d2^u + (1 - s_{t-1}[k]) * d3^u \ge z_{t-1}[k];$

            $Am_{t-1}[k] = \alpha * (Am_t^-[k] * d2^u + Am_t^+[k] * d2^l);$

            $tmp_s \ \ \ \ = \alpha * (Am_t^-[k] * d3^u + Am_t^+[k] * d3^l);$

            $As_{t-1}[k] - = tmp_s; \ \ \ bias + = sum(tmp_s);$

        **end if**

        //spatial update;

        $As_t[k-1] = Am_t[k] * w[k-1]; \ \ \ bias + = Am_t[k] * b[k-1];$

    **end for**

**end for**

**Return** $\sum_t \left( As_t[0] * (\dot{x}_t^u + \dot{x}_t^l)/2 - |As_t[0]| * (\dot{x}_t^u - \dot{x}_t^l)/2 \right) + bias$

---

# 4 Experiment

## 4.1 Experiment Setup

**Dataset and Network Structure**: In this work, we evaluate our robust training method on three datasets: MNIST [18], FMNIST [32] and NMNIST [24]. MNIST and FMNIST are digital datasets and NMNIST is spike dataset. For each dataset, we use two network structures in experiments. The detailed setting for datasets and network structures are shown in Table 1. We set the firing threshold $th$ and decay factor $\alpha$ to 0.25 in Eq. (1) and (3). Our experiments are conducted by Pytorch 1.8. The hardware we used is one Nvidia RTX3090 GPU and one AMD Ryzen CPU.

Table 1: Datasets and network structure

|  | Input Type | Size | Time Step | FC Acc | CONV Acc |
|---|---|---|---|---|---|
| MNIST | digital | 1*28*28 | 10 | 98.45% | 99.09% |
| FMNIST | digital | 1*28*28 | 10 | 87.58% | 89.53% |
| NMNIST | spike | 2*34*34 | 10 | 98.30% | 99.05% |
| FC | X-FC512-FC256-FC10 | | | | |
| CONV | X-C64K3S2-C128K3S2-FC256-FC10 | | | | |

**Original and Robust Training**: In the original training, we adopt BPTT based training [31]. We train 80 epochs for each SNN model. The learning rate is set to 0.01 at the beginning, it decays to 0.001 at the $55^{th}$ epoch. In robust training, we use the lower bound of S-CROWN as the loss function. During robust training, we set $\epsilon$ to 0 at the beginning. It will increase linearly to the final $\epsilon$ during the first 250 training epochs. In the last 50 training epochs, $\epsilon$ is unchanged.

**Adversarial Attack**: In this work we adopt the untargeted white-box gradient-based attack [19] in SNN. In our experiment, we select 300 examples for each dataset to apply adversarial attack. In order to bound the noise of adversarial examples, we involve additional constraints during the attack. Specifically, for digital inputs, we clip the adversarial example for each attack iteration to make the adversarial example stay in the boundary. For spike inputs, each attack iteration we limit the amount of changed elements to $size(\hat{x}) * \epsilon/2$ and $size(\hat{x}) * \epsilon/6$ for FC and CONV networks to achieve the highest attack efficiency.

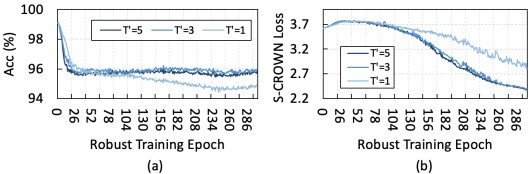

Figure 4: Impact of robust training time steps $T'$ on (a) original accuracy and (b) S-CROWN loss. (CONV model; MNIST dataset; $\epsilon = 0.12$)

## 4.2 Robust Training in SNNs

Table 2: Comparing untargeted white-box gradient-based attack between the original model and the model after robustness training.

|  | attack $\epsilon$ | MNIST (FC-$\epsilon$=0.12, CONV-$\epsilon$=0.12) original network org err | attack err | robust network org err | attack err | attack $\epsilon$ | FMNIST (FC-$\epsilon$=0.01, CONV-$\epsilon$=0.005) original network org err | attack err | robust network org err | attack err | attack $\epsilon$ | NMNIST (FC-$\epsilon$=0.005, CONV-$\epsilon$=0.005) original network org err | attack err | robust network org err | attack err |
|---|---|---|---|---|---|---|---|---|---|---|---|---|---|---|---|
| FC | 0.104 | 0.3% | 19.3% | 4.7% | 16.6% (-2.7%) | 0.040 | 13.0% | 20.7% | 13.6% | 20.0%(-0.7%) | 0.0007 | 3.7% | 21.0% | 3.7% | 10.0%(-11.0%) |
|  | 0.124 | 0.3% | 29.7% | 4.7% | 20.3% (-9.4%) | 0.070 | 13.0% | 29.7% | 13.3% | 24.0%(-5.7%) | 0.0009 | 3.7% | 30.7% | 3.7% | 12.3%(-18.4%) |
|  | 0.140 | 0.3% | 39.0% | 4.7% | 23.3%(-15.7%) | 0.100 | 11.7% | 41.3% | 14.0% | 32.0%(-9.3%) | 0.0012 | 3.7% | 41.3% | 3.7% | 16.7%(-24.6%) |
|  | 0.154 | 0.3% | 50.0% | 5.0% | 24.3%(-25.7%) | 0.114 | 13.3% | 49.7% | 13.6% | 38.7%(-11.0%) | 0.0014 | 3.7% | 48.7% | 3.7% | 18.3%(-30.4%) |
| CONV | 0.120 | 0.3% | 18.7% | 3.7% | 8.3%(-10.4%) | 0.040 | 10.0% | 19.7% | 18.0% | 22.0%(+0.3%) | 0.0008 | 1.3% | 20.0% | 4.3% | 7.0%(-13.0%) |
|  | 0.140 | 0.3% | 29.0% | 4.0% | 9.0%(-20.0%) | 0.060 | 10.3% | 28.3% | 17.3% | 23.0%(-5.3%) | 0.0010 | 1.3% | 33.7% | 4.3% | 11.0%(-22.7%) |
|  | 0.170 | 0.3% | 40.7% | 3.3% | 10.7%(-30.0%) | 0.080 | 10.0% | 41.3% | 17.0% | 25.7%(-15.6%) | 0.0012 | 1.3% | 41.0% | 4.3% | 11.0%(-30.0%) |
|  | 0.190 | 0.3% | 50.7% | 4.0% | 13.0%(-37.7%) | 0.100 | 10.3% | 52.3% | 16.7% | 34.3%(-18.0%) | 0.0015 | 1.3% | 49.0% | 4.3% | 12.0%(-37.0%) |

**Selection of Robust Training Time Steps** $T'$: We do not follow the original time steps during the robust training. Because the spatial computations in SNNs share the same parameters between different time steps, we can use arbitrary robust training time steps $T'$. In Figure 4 we analyze the impact of $T'$ on original accuracy and S-CROWN loss. Note that we would like to keep a higher original accuracy but reduce the S-CROWN loss after the robust training. From the result, we can find that $T' = 3$ gives the optimal solution. It implies that a too small $T'$ cannot capture the temporal

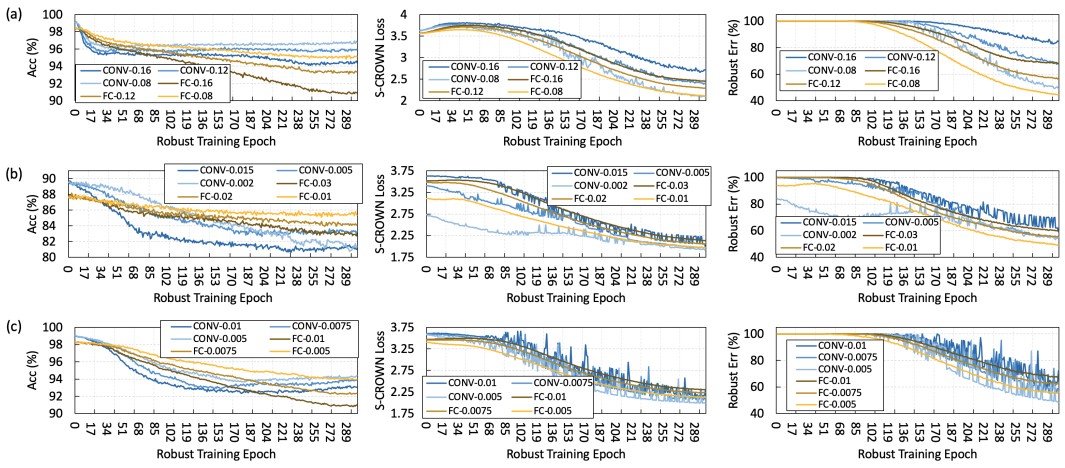

Figure 5: Original accuracy, S-CROWN loss and S-CROWN robust error during robust training under (a) MNIST; (b) FMNIST and (C) NMNIST datasets with different network structures and $\epsilon$.

dynamics of SNNs and a larger $T'$ may cause the boundary functions in S-CROWN to become too loose. Also we find that the robust training time for each epoch is $2.6\times$, $7.7\times$, and $12.9\times$ with respect to the original plain training when we set $T'$ to 1, 3, and 5. Thus, the selection of $T'$ also affects the robust training efficiency. In the rest of our evaluations, we set $T' = 3$ for all robust training.

**Robust Training on Variaous Datasets**: The analysis of robust training on different datasets with various $\epsilon$ and network structures is shown in Figure 5. From the result, we have the following observations: 1. During the robust training, it is more stable for a fully connected network(yellow curves are smoother than blue curves) because of the simpler network structure. 2. With a larger $\epsilon$, the original accuracy is dropped more after the robust training. Also, it is more difficult to achieve a smaller robust error with larger $\epsilon$. 3. In the MNIST dataset, when the network structure is CONV, the robust training is more stable as $\epsilon$ increases. The potential reason is that when $\epsilon$ is small, the flipped regions in spike inputs are more diverse. 4. In FMNIST and NMNIST datasets, when the network structure is CONV, the robust training is much more fluctuating. For FMNIST dataset, the unstable may be caused by the more complicated input data (cloths) and the lower convergence in original training (final accuracy is 89.53%). For NMNIST dataset, the unstable may come from the larger input data size. A larger input size indicates the potential input regions that can be attacked become larger. By considering the original accuracy and S-CROWN result after the robust training, we select the $\epsilon$ as shown in Table 2. We use FC-$\epsilon$ and CONV-$\epsilon$ to represent the noise boundary we selected for different network structures during robust training.

### 4.3 Network Robustness Evaluation

We use untargeted white-box adversarial attack to compare the robustness between the original model and the model after robust training. The robustness comparison is shown in Table 2. We use the original error rate to reflect the accuracy of a model on the 300 test data. The original error may have variance during the evaluation for the digital image dataset because of the input sampling mechanism. In our experiment, we select the attack $\epsilon$ to achieve approximately 20%, 30%, 40%, and 50% attack error rate on the original model. From the result, we can find that after the robust training, the models are harder to be attacked with adversarial example in all cases. We also find that the model after robust training is more secure when the attack $\epsilon$ is larger, even though the $\epsilon$ for robust training is much smaller. The potential reason is that the binary behavior of the spike events causes the boundary propagation to diverge quickly, which makes the final boundary cover more noisy inputs. Finally, we find that model robustness can be improved more when the network structure is CONV, since more parameters can be adjusted under the CONV model. From the result, we find that the CONV model in MNIST achieves the highest robustness improvement, i.e., the attack error rate reduces 37.7% with 3.7% original accuracy loss when the attack $\epsilon$ equals 0.190.

# 5 Discussion & Conclusion

In this work, we aim to design an efficient robust training method for SNNs based on the certified methods. Specifically, we design S-IBP and S-CROWN to tackle the distinct non-linear neuron behaviors in SNNs. Also, we formulate the input boundary for different input types. Based on our results, we can achieve at most $37.7\%$ attack error reduction with $3.7\%$ original accuracy loss, which demonstrates the efficiency of our proposed method.

## Border Impact

SNN is a class of neural networks that inherit the biological model of the neuron dynamics in the brain. As the accuracy of SNNs becomes higher, the security issue attracts more attention. In this work, we design S-IBP and S-CROWN that apply certification-based robust training to improve the robustness of an SNN model. Our work can help to improve the misclassification rate of an SNN in real-world deployments. We believe our work can provide a positive impact on the performance of the SNN models. Also, we encourage researchers to seek other techniques and factors that can potentially affect the performance of an SNN model.

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
