# Appendix

## A    S-CROWN on Directed Encoding

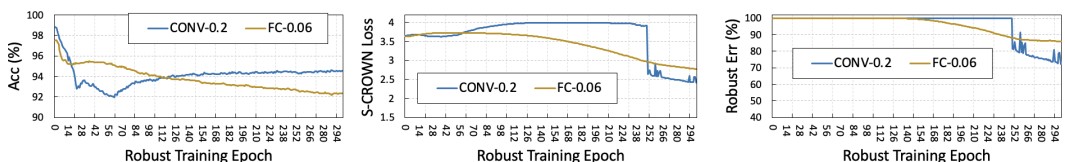

Figure 6: Original accuracy, S-CROWN loss and S-CROWN robust error during robust training under MNIST database with direct encoding.

Instead of rate-coding, directed encoding is another popular input coding mechanism that directly applies a CONV or FC layer on the original digital as the first layer [31, 16]. Currently, the directed encoding only takes digital inputs. Since the input does not involve sampling or binary patterns, we can apply $\ell_\infty$ norm to build the input bounds. The results of certification-based robust training are shown in Figure 6. From the result, we can find that the indicators (original acc, S-CROWN loss, S-CROWN robust error) during training follow a similar trend as the rate-coding.

Table 3: Untargeted white-box gradient-based attack on MNIST database with directed encoding.

|  | attack | original network | | robust network | |
|---|---|---|---|---|---|
|  | $\epsilon$ | org err | attack err | org err | attack err |
| FC | 0.060 | 1.3% | 21.0% | 7.7% | 22.7% (+1.7%) |
|  | 0.070 | 1.3% | 30.3% | 7.7% | 24.3% (-6.0%) |
|  | 0.080 | 1.3% | 38.0% | 7.7% | 26.3% (-15.7%) |
|  | 0.092 | 1.3% | 49.0% | 7.7% | 30.7% (-18.3%) |
| CONV | 0.150 | 0.0% | 21.3% | 7.3% | 11.7% (-9.6%) |
|  | 0.180 | 0.0% | 33.3% | 7.3% | 13.3% (-20.0%) |
|  | 0.200 | 0.0% | 42.0% | 7.3% | 15.0% (-27.0%) |
|  | 0.210 | 0.0% | 50.3% | 7.3% | 20.7% (-29.6%) |

*MNIST (FC-$\epsilon$=0.12 CONV-$\epsilon$=0.20)*

Next, we evaluate the model robustness against adversarial training in Table 3. As the result, our method can at most achieve 29.6% error rate reduced with 7.3% original accuracy loss, which demonstrated the effectiveness of our proposed method on directed encoding.

## B    Example of Input Bounds

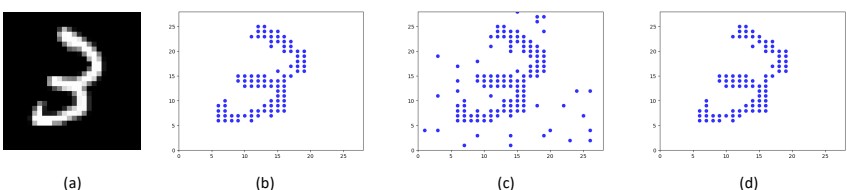

|  (a)  |  (b)  |  (c)  |  (d)  |

Figure 7: A visual example of bounds for one time step in the MNIST database: (a) the original digital image; (b) the sampled result based on the original image (c) the sampled upper bound and (d) the sampled lower bound.

An example of input bounds after sampling for the MNIST database is shown in Figure 7. Here, we show the upper and lower bounds for one time step. After we applied the bounding method in Section 3.2 the background of the lower bound image before sampling is the same as the original image whose gray values are 0. In contrast, the skeleton of the content in the upper bound image before sampling is unchanged compared to the original image. The gray values of the skeleton are 1. Note that the unstable points are located in the region where the gray values in the bounded image differ from the original image. Thus, the number of unstable points in the upper bound after sampling is more, most of them located in the background.