# OpenReview forum: "Toward Robust Spiking Neural Network Against Adversarial Perturbation"
_NeurIPS.cc/2022/Conference — NeurIPS 2022 Accept_

### Official Review · Reviewer_6QhU · 2022-06-30

**Rating:** 6
**Confidence:** 4
**Soundness:** 2 fair
**Presentation:** 2 fair
**Contribution:** 3 good

**Summary:**

In this paper, a robust training method for SNNs is proposed. It is based on S-IBP and S-CROWN algorithms. The results on 3 different datasets show attack error reduction with some original accuracy loss.

**Questions:**

1.	In Section 1, typo, rewrite the sentence: “Except for the biological character of SNN models, the event-driven based information propagation mechanism makes SNN can be deployed with limited computation resources.
2.	Is the methodology for generating adversarial attacks in SNN described in Section 2.2 novel, or is it inspired by related work? If the latter, the references should be included in the paragraph.
3.	Please describe in more detail which are the assumptions made to derive the equations in Section 3.1.
4.	In Section 3.2, typo, correct the sentence: “Also, the l0-norm boundary can be interpreted as the probability of an element is unstable”
5.	Section 3.3 only describes the S-IBP and S-CROWN algorithms independently. More details are required to describe the overall structure of the proposed robust training method.
6.	What is the tool flow used for conducting the experiments? Which packages/libraries are used?
7.	In Section 4.2: “By considering the original accuracy and S-CROWN result after the robust training, we select the ϵ as shown in Table 2.” It is not clear which criteria are used to select ϵ.
8.	How does this work compare with the following article?
Kundu, Souvik et al. “HIRE-SNN: Harnessing the Inherent Robustness of Energy-Efficient Deep Spiking Neural Networks by Training with Crafted Input Noise.” 2021 IEEE/CVF International Conference on Computer Vision (ICCV) (2021): 5189-5198.


**Limitations:**

The limitations and societal impact of this work have not been discussed. However, there is no reason to penalize the submission for this, since this work provides only positive impacts.

**Strengths And Weaknesses:**

Strengths:

1.	The contributions of this paper are clear and original.
2.	The achieved results are significant and relevant to advancing the state-of-the-art.

Weaknesses:

1.	There are several typos and semantically incorrect sentences throughout the text. It is recommended to conduct thorough proofreading.
2.	The clarity of some key sections can be improved. See comments below.

---

> ### Author Response · Authors · 2022-08-02
> **Dear reviewer 6QhU, Thank you for your careful check of our manuscript. We fixed typos and revised our paper based on your helpful suggestions.**
>
> ### 1. Rewrite the sentence: “Except for the biological character of SNN models, the event-driven based information propagation mechanism made SNN can be deployed with limited computation resources.”
>
> We have rewritten it as follows: “Among the rich biological characters of SNNs, the event-driven information propagation mechanism makes SNNs easy to deploy with limited computation resources.”
>
>  &nbsp;
>
> ### 2. Related work of adversarial attack.
>
> Our attack method is based on the reference below. We have included the reference in **Section 2.2**.
>
> *Liang, Ling, et al. "Exploring adversarial attack in spiking neural networks with spike-compatible gradient." IEEE transactions on neural networks and learning systems (2021).*
>
>  &nbsp;
>
> ### 3. Please describe in more detail which are the assumptions made to derive the equations in Section 3.1
>
> For each equation in S-IBP, we derive the S-IBP output bounds based on S-IBP input bounds. Thus, the assumption here is that we have acquired the S-IBP input bounds. Since the S-IBP is propagated forward, we can derive the bounds sequentially.
>
> For each equation in S-CROWN, we also need to know the S-IBP input bounds which can determine the S-CROWN boundary functions. Still, the assumption here is that we have acquired the S-IBP input bounds. Since the S-IBP runs before the S-CORWN, we have already acquired the S-IBP bound of each element during the S-CROWN phase.
>
> We have clarified the assumption in **Section 3.1**.
>
>  &nbsp;
>
> ### 4. In Section 3.2, typo, correct the sentence: “Also, the $l_0$-norm boundary can be interpreted as the probability of an element is unstable.”
>
> We have modified it: “Also, the $l_0$-norm boundary can be interpreted as the probability of an unstable element.”
>
>  &nbsp;
>
> ### 5. Section 3.3 only describes the S-IBP and S-CROWN algorithms independently. More details are required to describe the overall structure of the proposed robust training method.
>
> We have added an overall description in **Section 3.3**. For the overall flow, our method will first run S-IBP to get the boundary of intermediate data. Then we apply the S-CROWN which needs the S-IBP bound. Finally, we apply robust training (Zhang et al. 2014).
>
> *Zhang, Huan, et al. "Towards stable and efficient training of verifiably robust neural networks." arXiv preprint arXiv:1906.06316 (2019).*
>
>  &nbsp;
>
> ### 6. What is the tool flow used for conducting the experiments? Which packages/libraries are used?
>
> Our experiments are conducted by Pytorch 3.8. The hardware we use is GPU V100. We have clarified in **Section 4.1**.
>
>  &nbsp;
>
> ### 7. It is not clear which criteria are used to select $\epsilon$.
>
> In this work, $\epsilon$ is selected mainly based on the perturbation magnitude for the adversarial attack, i.e. the “attack $\epsilon$” in Table2. However, we still need additional empirical experiments to select a more efficient $\epsilon$ for the robust training. We have added some descriptions in **Section 3.2**.
> How to efficiently select $\epsilon$ is a very important topic in certification-based robust training. We will leave it in our future work.
>
>  &nbsp;
>
> ### 8. How does this work compare with HIRE-SNN.
>
> In HIRE-SNN, the robust training method is based on adversarial training, and the input perturbation during training is generated through adversarial attack. Compared to the HIRE-SNN, our certification-based robust training is based on the defined input boundary. Also, adversarial training and certification-based training are orthogonal to each other that can be combined. In ANNs, the combination of these two methods to improve NN model robustness is well-studied (Balunović et al. 2020; Fan et al. 2021). We have added some discussion in the **Section 2.3** and we will leave the further research in our future work.
>
> *Balunović, Mislav, and Martin Vechev. "Adversarial training and provable defenses: Bridging the gap." 8th International Conference on Learning Representations (ICLR 2020)(virtual). International Conference on Learning Representations, 2020.*
>
> *Fan, Jiameng, and Wenchao Li. "Adversarial Training and Provable Robustness: A Tale of Two Objectives." Proceedings of the AAAI Conference on Artificial Intelligence. Vol. 35. No. 8. 2021.*

---

> > ### Comment · Reviewer_6QhU · 2022-08-08
> > **Response to author rebuttal**
> >
> > Thank you for the detailed answers to the reviewers' comments. After the rebuttal and in light of all the discussions, I change my score to 6.

---

> > > ### Author Response · Authors · 2022-08-08
> > > **Response to Reviewer 6QhU**
> > >
> > > Thanks for your acknowledgment of our work.

---

### Official Review · Reviewer_2YgY · 2022-07-12

**Rating:** 4
**Confidence:** 4
**Soundness:** 2 fair
**Presentation:** 3 good
**Contribution:** 2 fair

**Summary:**

The paper proposes a methodology to tackle adversarial robustness in SNNs.

**Questions:**

Please see above comments.

**Limitations:**

Please see weakness section.

**Strengths And Weaknesses:**

+ The authors show that their method is able to resist attacks of different types on small scale datasets.
- The authors evaluation is pretty limited. In [1, 3], the authors show that BNTT trained SNN models are inherently more robust. Can the authors comment on how their methodology is different from [1,3]. Further, the authors started their paper discussion with SNNs being advatangeous on hardware, so it makes more sense to develop hardware aware robustness. But, the author's method is algorithm-based. Can the authors comment if their robustness will transfer to hadware as is or any modification will be required? In [4], the authors show that adversarial robustness on hardware become pretty low, so they come up with a normalization technique to resist attacks. In [2], the authors show that the type of coding technique plays a role in determining adversarial robustness. I am not sure if teh author's methodology can tarnsfer across differnet coding techniques.

[1] Revisiting batch normalization for training low-latency deep spiking neural networks from scratch Y Kim, P Panda Frontiers in neuroscience, 1638

[2] Rate Coding Or Direct Coding: Which One Is Better For Accurate, Robust, And Energy-Efficient Spiking Neural Networks? Y Kim, H Park, A Moitra, A Bhattacharjee, Y Venkatesha, P Panda ICASSP 2022-2022

[3] Visual explanations from spiking neural networks using interspike intervals Y Kim, P Panda Scientific Reports 11, Article number: 19037 (2021)

[4] Bhattacharjee, Abhiroop, et al. "Examining the Robustness of Spiking Neural Networks on Non-ideal Memristive Crossbars." arXiv preprint arXiv:2206.09599 (2022).

---

> ### Author Response · Authors · 2022-08-02
> **Dear reviewer  2YgY, Thank you for your constructive suggestions. We made comprehensive comparisons with related works you pointed out as follows and added the additional experiments as you suggested.**
>
> ### 1. Compare the proposed method with reference [1,3] and discuss how to transfer the proposed method to improve the SNN robustness on hardware (compare with the normalization method in [4]).
>
> References [1][3] find that the normalization can not only enhance the accuracy of an SNN model but also improve the robustness of SNN against input perturbation. Furthermore, reference [4] identifies that a noise-aware normalization method can also help to mitigate the impact of intrinsic crossbar non-idealities on accuracy drop. In general, the normalization method tries to improve the robustness by adjusting activation distribution and our proposed method considers this problem from the activation boundary perspective.
>
> Instead of identifying the differences, our proposed method is potentially orthogonal with other techniques that can improve the SNN accuracy against perturbation. Since all operations in normalization are linear equations, there is no conflict to integrate normalization into our method. We can adapt the well-studied methods (Tsuzuku et al. 2018; Wang et al. 2021) to design the corresponding boundary functions. After integrating the normalization, our method may be more robust against non-ideal hardware devices.
>
> It would be a very interesting and promising direction for future research after considering the normalization. Because of the rebuttal time budget, we will leave it in our future work. And we have added a discussion in **Section 2.3**.
>
> *Tsuzuku, Yusuke, Issei Sato, and Masashi Sugiyama. "Lipschitz-margin training: Scalable certification of perturbation invariance for deep neural networks." Advances in neural information processing systems 31 (2018).*
>
> *Wang, Shiqi, et al. "Beta-crown: Efficient bound propagation with per-neuron split constraints for neural network robustness verification." Advances in Neural Information Processing Systems 34 (2021): 29909-29921.*
>
>  &nbsp;
>
> ### 2. Transfer the proposed method to a different coding method i.e. the direct coding in [2].
>
> The coding method plays a vital role in the design of SNNs. Compared to rate coding, direct coding is even easier to bound, since it does not involve sampling on the original image. We can adopt a $l_{\infty}$ -norm on the input image.  For other coding methods like temporal coding and phase coding, since they also do not involve sampling or binary patterns in the input boundary, we can follow the well-studied $l_{\infty}$ -norm bounding method. We have added additional experiments in **Appendix A**. Based on our result, we can at most achieve 29.6\% error rate reduced with 7.3\% original accuracy loss on MNIST database, which demonstrated the effectiveness of our proposed method on directed encoding.

---

> > ### Comment · Reviewer_2YgY · 2022-08-08
> > **Rate coding effect**
> >
> > Thanks for the detailed response. In my opinion, coding is central to SNN robustness. What you have shown with rate coding may be a result of obfuscation which in my opinion might be a major benefactor to your bounding approach. I think the bounding with a rate coded input brings a stochastic behavior during the adversarial perturbation creation for the SNN that might be in turn resulting in robustness. I am still not convinced if this method will perform well with larger scale SNNs on natural image data (with other coding types).
> >
> >
> > I will keep my score as is.

---

> > > ### Author Response · Authors · 2022-08-08
> > > **Title: Dear Review 2YgY, thanks for your response.**
> > >
> > > For your concerns, we have the following comments: Firstly, rate coding is one of the most important coding mechanisms for SNN. As we all know, the DVS devices can generate the input with rate coding directly. Thus, in this work, we mainly discuss our method on rate coding. Secondly, for other coding mechanisms, i.e. direct coding, we have demonstrated the effectiveness of our method in **Appendix A**. At last, our method is a fusible method that can be integrated with other approaches to improve the robustness of a model together.
> > >
> > > As a pioneer study on verification-based robustness training, we have demonstrated the effectiveness of our method.  Since applying verification-based technologies to Neural Networks is one of the most rigorous and hardest tasks in adversarial robustness evaluation, we cannot include all aspects in one work.  It would be very promising to keep investigating the directions you suggested, i.e. larger model and various coding mechanisms.

---

### Official Review · Reviewer_d88H · 2022-07-13

**Rating:** 8
**Confidence:** 5
**Soundness:** 4 excellent
**Presentation:** 3 good
**Contribution:** 3 good

**Summary:**

This paper considers that spiking neural networks are not suitable for traditional adversarial robustness analysis methods, and proposes linear relaxations for the membrane potential and spikes of SNNs. These relaxations can be used to provide robust training for SNNs.

**Questions:**

Is this the first analysis of the robustness of spiking neural networks? [1] is also a paper that copes with the robustness of SNNs.

[1] HIRE-SNN: Harnessing the Inherent Robustness of Energy-Efficient Deep Spiking Neural Networks by Training With Crafted Input Noise



**Limitations:**

yes

**Strengths And Weaknesses:**

Pros:

This paper is well written.

This paper gives a linear relaxation scheme for SNNs.

This solution takes into account not only temporal updates but also spatial updates.



Cons:

I'm not sure if the linear relaxation is too loose. I think an example of MNIST can be used to illustrate the gap.

The upper and lower bounding strategies used for spike inputs in Sec 3.2 are probabilistic, not linear. Can the introduction of probability be consistent with linear relaxation? If not, i.e. x^u=x^i=x, how will the model work?

Previous work has shown the robustness of SNN is affected by coding (Poisson coding and directly coding) [1]. What's the way of coding in your paper?

There are some typos that need to be fixed. And the citation is not standardized. For example: "leaky integrated-and-fire (LIF) Gerstner et al. (2014)”（Line 56）should be “leaky integrated-and-fire (LIF) (Gerstner et al. 2014)”.

[1] HIRE-SNN: Harnessing the Inherent Robustness of Energy-Efficient Deep Spiking Neural Networks by Training With Crafted Input Noise

---

> ### Author Response · Authors · 2022-08-02
> **Dear Reviewer d88H, Thank you for your valuable comments. We addressed your concerns in the rebuttal as follows**
>
> ### 1. Show the relaxation through MNIST examples.
>
> We provide visual examples for the upper and lower bounds of MNIST and NMNIST datasets in **Appendix B**.
>
> &nbsp;
>
> ### 2. The bounds in Sec 3.2 are probabilistic. Can the introduction of probability be consistent with linear relaxation? If not, i.e. $x^u=x^l=x$, how will the model work?
>
> In our method, we can first guarantee $s^u \geq s^l$ for each element $s$ in a (sampled) spike input. Then, our method can provide $N * \epsilon$ unstable elements that satisfy corresponding $s^u=1$ and $s^l=0$ from the probabilistic perspective. Here, $N$ is the total number of elements in the (sampled) spike input and $\epsilon$ is the boundary distance. Usually, the total number of elements for a (sampled) spike input is large, i.e. 10*28*28 elements for the MNIST database (10 is the timesteps in our experiments).  We need to avoid a too small $\epsilon$ to make the upper and lower bounds identical i.e. for MNIST, we can get one unstable element once the selected $\epsilon$ is greater than 0.0013 theoretically. Once there are some unstable elements, our bounds for (sampled) spike inputs are consistent with linear relaxation. We have clarified the relation of our approach with linear relaxation in **Section 3.2**.
>
> Another possible approach is to randomly select a certain portion of elements in a (sampled) spike input to be unstable elements, which can make $X^u \ne X^l$. We will discuss it in our future work.
>
> &nbsp;
>
> ### 3. Some typos that need to be fixed. And the citation is not standardized.
>
> We have fixed typos and standardized citations in our revised version.
>
> &nbsp;
>
> ### 4. Compare with HIRE-SNN
>
> In HIRE-SNN, the robust training method is based on adversarial training, and the input perturbation during training is generated through adversarial attack. Compared to the HIRE-SNN, our certification-based robust training is based on the defined input boundary. Also, adversarial training and certification-based training are orthogonal to each other that can be combined. In ANNs, the combination of these two methods to improve NN model robustness is well-studied (Balunović et al. 2020; Fan et al. 2021). We have added some discussion in the **Section 2.3** and we will leave the further research in our future work.
>
> *Balunović, Mislav, and Martin Vechev. "Adversarial training and provable defenses: Bridging the gap." 8th International Conference on Learning Representations (ICLR 2020)(virtual). International Conference on Learning Representations, 2020.*
>
> *Fan, Jiameng, and Wenchao Li. "Adversarial Training and Provable Robustness: A Tale of Two Objectives." Proceedings of the AAAI Conference on Artificial Intelligence. Vol. 35. No. 8. 2021.*

---

> > ### Comment · Reviewer_d88H · 2022-08-07
> > **Responses**
> >
> > Thanks for the rebuttal. Most of my concerns are appropriately addressed. However, I recommend adding more discussion on hyper-parameter settings. Considering this is the first work on certification-based robust training of SNNs and the potential ability to integrate it into other techniques, I'd like to raise my rating score to 8.

---

> > > ### Author Response · Authors · 2022-08-08
> > > **Response to Reviewer d88H**
> > >
> > > Thanks for your support of our work. We will provide more detail on hyper-parameter settings in our next version.

---

### Meta-Review · Area_Chair_QWCb · 2022-08-28

**Recommendation:** Accept
**Confidence:** Certain

**Metareview:**

This paper applies existing certification-based adversarial robustness techniques to spiking neural networks. They achieve this through upper and lower relaxations of the spiking equations.

Review scores were high variance, ranging from 4 through 8. Reviews were generally of high quality. The largest concern was that the use of rate coding for the network's output limited the applicability of the technique. I found the authors' response to this concern satisfying.

I appreciate that this paper is the first to apply certification-based techniques to spiking neural networks. I believe it has the potential to produce significant impact for that reason.

Based upon the reviews, and my judgement of the potential impact, I recommend the paper be accepted.

**Award:**

No

---

### Decision · Program_Chairs · 2022-09-14

Accept